# EquiBot: SIM(3)-Equivariant Diffusion Policy for Generalizable and Data Efficient Learning

**Jingyun Yang**\*, **Zi-ang Cao**\*, **Congyue Deng**,
**Rika Antonova, Shuran Song, Jeannette Bohg**
Stanford University
{jingyuny, ziangcao, congyue, rika.antonova, shuran, bohg}@stanford.edu

Figure 1: We propose a method for learning **generalizable** and **sample-efficient** visuomotor policies that can be applied to **everyday manipulation tasks**.

**Abstract:** Building effective imitation learning methods that enable robots to learn from limited data and still generalize across diverse real-world environments is a long-standing problem in robot learning. We propose EquiBot, a robust, data-efficient, and generalizable approach for robot manipulation task learning. Our approach combines SIM(3)-equivariant neural network architectures with diffusion models. This ensures that our learned policies are invariant to changes in scale, rotation, and translation, enhancing their applicability to unseen environments while retaining the benefits of diffusion-based policy learning such as multi-modality and robustness. We show on a suite of 6 simulation tasks that our proposed method reduces the data requirements and improves generalization to novel scenarios. In the real world, with 10 variations of 6 mobile manipulation tasks, we show that our method can easily generalize to novel objects and scenes after learning from just 5 minutes of human demonstrations in each task. Website: https://equi-bot.github.io/

**Keywords:** Imitation Learning, Equivariance, Data Efficiency

## 1 Introduction

The unpredictability and variability of real-world environments have posed significant challenges to the development of fully autonomous robotic agents. Existing visuomotor policies learned via imitation learning are effective in controlled settings [1, 2, 3, 4, 5, 6, 7], but they require substantial data and generalize poorly to unseen scenarios [8, 9, 10, 11, 12], limiting their practical deployment.

In this work, we introduce EquiBot, an equivariant policy learning architecture based on diffusion models [13]. The equivariant neural network guarantees that outputs scale, translate, and rotate with inputs, even if not fully trained [14], enabling generalization to unseen scenarios with changes in

---

\* These authors contributed equally.

8th Conference on Robot Learning (CoRL 2024), Munich, Germany.

object visuals, sizes, and placements. This architecture also has data efficiency benefits, as it can infer actions for object placements and poses that are in distribution but insufficiently demonstrated, outperforming non-equivariant policies. Using a diffusion model-based policy architecture also offers robust learning performance, including support for idle actions and multi-modal behaviors [8]. Leveraging this design, EquiBot enables a wide range of household manipulation tasks using only a handful of single-view human demonstration videos. At test time, our method takes single-view, object-centric point clouds and robot proprioception as input and outputs sequences of robot end-effector actions that consist of 6D velocity commands and gripper open-close signals.

We evaluate our method in both simulation and real-world experiments. In simulation, we test on a suite of six tasks adapted from prior benchmarks [8, 15, 16], showing that our method is more data-efficient and generalizes better to unseen scenarios than prior works. In real robot experiments, we quantitatively test on 6 real-world mobile manipulation tasks in home settings, including pushing a chair towards a desk, closing a laundry machine door, folding towels, making a bed, and closing a suitcase. Our method successfully performs these tasks with unseen objects and scenes from just 5 minutes of human demonstrations, outperforming baselines that rely on augmentations for generalization or do not use a diffusion process to predict actions.

## 2    Related Work

**Data-efficient imitation learning.** Recent imitation learning approaches for imitation learning assume large quantities of data to be available for learning manipulation policies [4, 8]. To reduce the amount of data that policy training requires per task, some work [2, 17, 18, 19] formulate a one-shot or few-shot imitation learning setup where the policy is trained on multi-task demonstration data and can then output actions in a novel task after seeing one or more demonstrations as well as the current state. However, this approach requires the availability of multi-task data in a task domain. Some other works [20] achieve imitation learning from small demonstration datasets with sampling-based optimization methods like Bayesian Optimization, but these methods are often limited to small action spaces and open-loop settings. In contrast to prior works, we show that embedding equivariance into the policy architecture can effectively improve the data efficiency of the imitation learning algorithm, allowing robust policies to be learned with just a handful of demonstrations.

**Equivariance in robot manipulation.** To help learned robot policies generalize to unseen environments and object placements, prior works [21, 22, 23] use data augmentation to improve cross-domain transfer of the learned policy. However, these methods increase training time significantly and do not guarantee generalization to visual appearances, object scales, and poses that are unseen in the training data distribution even after the augmentation. In contrast, utilizing equivariant representations in policy learning allows the learned policies to generalize to objects and initial conditions not previously seen at training time. Prior works have explored the use of equivariance in robot manipulation in several different setups [24, 25, 26, 27, 28, 29], but most of them either focus on only simple pick-and-place like tasks, do not support closed-loop policies, or do not support scale equivariance. In our prior work [16], we developed a SIM(3)-equivariant visuomotor policy learning method that can go beyond pick-and-place tasks with deformable and articulated objects. However, EquivAct cannot handle multi-modal training data because of its deterministic architecture. Equiv-Act also lacks the necessary architectural design that prevent it from predicting multiple actions into the future, which is crucial for producing temporally consistent actions [8]. Compared to prior works, our method trains stably, handles multi-modal data with diffusion, and generalizes to unseen object appearance, initial states, scales, and poses with SIM(3)-equivariance.

**Equivariant diffusion architectures.** Prior works have integrated equivariance in diffusion models in various non-robotics domains [30, 31, 32, 33]. Some works have attempted to integrate equivariant architectures in diffusion models for robotics [34, 35, 36]. Diffusion-EDFs [34] produces target end-effector poses and can only solve pick-and-place tasks. EquiDiff [35] can only handle SO(2)-equivariance with simple 2D trajectories. EDGI [36] assumes ground-truth scene states as input

and cannot handle complex visual observations. Compared to prior works, our work proposes an equivariant diffusion policy for closed-loop 3D manipulation tasks with point cloud observations.

## 3 Method

In this section, we describe the design of EquiBot. Our method builds upon recent advances that use a diffusion process to represent visuomotor policies [8]. Starting from that, we provide key insights for injecting equivariant architectures. By building an equivariant noise prediction network, we enforce each diffusion step to be equivariant by construction (see Figure 2), and because of the self-symmetry of the initial Gaussian noise, the overall framework outputs an equivariant distribution under stochasticity. Section 3.2 provides our formal argument for this. The equivariant update in the diffusion process plays the role of letting the network "see" different input variations under transformations, which replaces the need for data augmentations and makes our framework data-efficient. Below, we introduce concepts related to equivariance and diffusion policy, then describe our method.

### 3.1 Preliminaries

**Problem setup.** We assume an imitation learning setup, where our method receives a demonstration dataset $\mathcal{D} = \{\tau^n\}_{n=1}^N$, which consists of $N$ demonstration trajectories $\tau^n$. Each demonstration trajectory consists of sequences of observation-action pairs $(\mathbf{O}_t, \mathbf{A}_t)$. The goal of the policy is to learn a mapping $\pi$ from past observations $\mathbf{O}_{t-T_o:t}$ to either the next action $\mathbf{A}_t$ or the next set of actions $\mathbf{A}_{t:t+T_p}$, where $T_o$ and $T_p$ are the observation and prediction horizons. At evaluation time, the policy receives state $\mathbf{O}_t$ and predicts the next one or more actions to be executed in the environment. In this work, we assume the observation $\mathbf{O}_t = (\mathbf{X}_t, \mathbf{S}_t)$ is composed of the scene point cloud $\mathbf{X}_t$ and robot proprioception $\mathbf{S}_t$.

**SIM(3)-equivariant network architectures.** Let $f$ be a function that takes a point cloud

Figure 2: **Method overview.** Given input scene point cloud & robot pose, our method performs a series of diffusion steps to obtain denoised actions with SIM(3)-equivariance, i.e. when the inputs translate, rotate, and scale, the outputs are guaranteed to translate, rotate, and scale accordingly.

$\mathbf{X} \in \mathbb{R}^{N \times 3}$ as input. This function is considered SIM(3)-equivariant if $f(\mathbf{TX}) = \mathbf{T}f(\mathbf{X})$ for any rigid 3D transformation $\mathbf{T} := (\mathbf{R}, \mathbf{t}, s) \in \text{SIM}(3)$, where $\mathbf{R}$, $\mathbf{t}$, and $s$ denote rotation, translation, and scale respectively. In this work, we use the same SIM(3)-equivariant encoder architecture and network layers as [16].

**Diffusion process as policy representation.** Our method uses Denoising Diffusion Probabilistic Models (DDPMs) to model the conditional distribution $p(\mathbf{A}_t | \mathbf{O}_t)$ similar to [8]. Starting from Gaussian noise $\mathbf{A}_t^K$, where $K$ is the number of diffusion steps, DDPM performs $K$ iterations of denoising to predict actions with decreasing levels of noise, $\mathbf{A}_t^{K-1}, \ldots, \mathbf{A}_t^0$. This process follows

$$\mathbf{A}_t^{k-1} = \alpha_k(\mathbf{A}_t^k - \gamma_k \epsilon_\theta(\mathbf{O}_t, \mathbf{A}_t^k, k) + \sigma_k \mathcal{N}(0, \mathbf{I})), \tag{1}$$

where $\epsilon_\theta$ is a denoising network, $\mathcal{N}(0, \mathbf{I})$ is Gaussian noise, and $\alpha_k, \gamma_k, \sigma_k$ are functions of $k$ set by a noise scheduler. The policy outputs $\mathbf{A}_t^0$ as its inference output. In this work, we use the CNN-based Diffusion Policy variant specified in [8] as the starting point of our architecture design. The original CNN-based Diffusion Policy architecture uses a noise prediction network that takes observation $\mathbf{O}_t$, diffusion iteration $k$, and noisy action $\mathbf{A}_t$ as input, and predicts the gradient $\nabla \mathbf{E}(\mathbf{A}_t)$ for denoising $\mathbf{A}_t$. The network first uses an encoder to encode the visual observations. The encoded visual features and positional embeddings of the diffusion iteration parameter are passed into FiLM layers [37] so that the encoded visual inputs are integrated into the network. Then, the policy network uses a

convolutional U-net [38] to process the input noisy actions $\mathbf{A}_t$, the conditioned observations, and diffusion iteration $k$ to predict the output denoising gradients.

**Observation and action spaces.** We use point clouds as input observations, since these contain the necessary 3D information to ensure policies can be structured as equivariant to translation, rotation, and scaling. We represent robot proprioception information with $\mathbf{S}_t = (\mathbf{S}_t^{(x)}, \mathbf{S}_t^{(d)}, \mathbf{S}_t^{(s)})$, with 3D positions in $\mathbf{S}_t^{(x)}$, normalized directions in $\mathbf{S}_t^{(d)}$, and scalars in $\mathbf{S}_t^{(s)}$. Robot proprioception can be converted into such a format that uses positions, velocities, offsets, and scalars in most cases, e.g. end-effector positions go to $\mathbf{S}_t^{(x)}$; end-effector velocities can be converted to position targets and go to $\mathbf{S}_t^{(x)}$; end-effector orientations can be converted into rotation matrices and placed in $\mathbf{S}_t^{(v)}$; gripper open-close states go to $\mathbf{S}_t^{(s)}$. Similarly, our representation for actions $\mathbf{A}_t = (\mathbf{A}_t^{(v)}, \mathbf{A}_t^{(d)}, \mathbf{A}_t^{(s)})$ consists of 3D offsets or velocities $\mathbf{A}_t^{(v)}$, normalized directions $\mathbf{A}_t^{(d)}$, and scalars $\mathbf{S}_t^{(s)}$. Similar to proprioception information, most existing action spaces can also be converted into this format.

## 3.2 Equivariant Distributions and Diffusion

We make a diffusion process equivariant by making the architecture for performing each diffusion step equivariant. In this section, we show that with equivariance in the per-step diffusion update, the final output action distribution is also equivariant under stochasticity. We mainly discuss equivariance to SO(3)-rotations in the diffusion process. Equivariance to translations and scaling (to get SIM(3)-equivariant architectures) is achieved via canonicalization before the diffusion process.

**Proposition 1.** *Let $p(x^K|c)$ be an SO(3)-equivariant density function conditioned on $c$, i.e. $\forall \mathbf{R} \in SO(3), p(x^K|c) = p(\mathbf{R}x^K|\mathbf{R}c)$. If the Markov transitions $p(x^{k-1}|x^k, c)$ are SO(3)-equivariant for all $k$, i.e. $p(x^{k-1}|x^k, c) = p(\mathbf{R}x^{k-1}|\mathbf{R}x^k, \mathbf{R}c)$, then the density $p(x^0|c) = \int p(x^T|c)\Pi_{k=1}^K p(x^{k-1}|x^k, c)$ is also SO(3)-equivariant.*

Please view the complete proof in Section D of the supplementary materials. In our case, observation $\mathbf{O}_t$ conditions the diffusion process of the actions $\mathbf{A}_t^{0:K}$. The prior distribution $p(\mathbf{A}_t^K|\mathbf{O}_t)$ is a standard Gaussian distribution equivariant to SO(3) transformations. The transition probabilities $p(\mathbf{A}_t^{k-1}|\mathbf{A}_t^k, \mathbf{O}_t)$ are predictions by an equivariant network, and thus are equivariant to SO(3)-rotations. Therefore, the final output action $p(\mathbf{A}_t^0|\mathbf{O}_t)$ is also SO(3)-equivariant.

## 3.3 SIM(3)-Equivariant Diffusion Policy

To design a SIM(3)-equivariant model architecture, our approach is to modify each part of the CNN-based Diffusion Policy architecture [8] to make them individually equivariant architectures. First, we design our point cloud encoder to be SIM(3)-equivariant and additionally output a centroid vector $\Theta_c$ as well as a scalar $\Theta_s$ quantifying object scale. $\Theta_c$ and $\Theta_s$ are then used to scale the inputs to subsequent layers of the network so that they are invariant to positions and scales. We then modify the FiLM layers, the convolutional U-net architecture, and other connecting layers to be SO(3)-equivariant. Finally, before producing the output actions, we scale relevant parts of the action back using $\Theta_c$ and $\Theta_s$ so the output is SIM(3)-equivariant to the input observation. Please see supplementary materials (Figure 13) for a detailed method figure.

**Encoder.** We use a PointNet-based [39] encoder in this work. For SIM(3)-equivariance, we reuse the encoder $\Phi$ introduced in [40] and [16]. This encoder takes a point cloud $\mathbf{X}$ as input and outputs a latent code $\Theta = \Phi(\mathbf{X})$, comprised of four components: $\Theta := (\Theta_R, \Theta_{\text{inv}}, \Theta_c, \Theta_s)$, where $\Theta_R$ is a rotation equivariant latent representation, $\Theta_{\text{inv}}$ is an invariant latent representation, scalar $\Theta_s$ is the computed object scale, and vector $\Theta_c$ denotes the object centroid. For more details on this encoder, we refer to [16]. While [16] pre-trains the encoder using generated simulation data, we do not perform pre-training on the encoder and learn it from scratch. This eliminates the need to build task-specific simulation environments and collect custom pre-training data in these environments.

**Routing input observations and actions into a conditional U-net.** The conditional U-net takes two inputs: action representation $\mathbf{Z}_a$ and conditioning information $\mathbf{Z}_c$. To construct these inputs

from point cloud encoding $\Theta$, proprioception $\mathbf{S}_t$, and noisy action $\mathbf{A}_t$, we need to first translate and scale $\mathbf{S}_t$ and $\mathbf{A}_t$ using $\Theta_c$ and $\Theta_s$ so the resulting values are invariant to scale and position, and then merge relevant inputs. More concretely, we define the action representation as

$$\mathbf{Z}_a = f_{\text{fuse}}\big([\mathbf{A}_t^{(v)}/\Theta_s, \mathbf{A}_t^{(d)}], \mathbf{A}_t^{(s)}\big), \qquad (2)$$

where $f_{\text{fuse}}$ is a Vector Neuron layer that takes vector information as its first and scalar information as its second input argument. We define conditioning information as

$$\mathbf{Z}_c = (\mathbf{Z}_c^{(\text{vector})}, \mathbf{Z}_c^{(\text{scalar})}) = \Big([\Theta_{\text{inv}}, (\mathbf{S}_t^{(x)} - \Theta_c)/\Theta_s, \mathbf{S}_t^{(d)}], [\mathbf{S}_t^{(s)}, \texttt{pos\_emb}(k)]\Big), \qquad (3)$$

where $\mathbf{Z}_c^{\text{vector}}$ and $\mathbf{Z}_c^{\text{scalar}}$ are vector and scalar conditioning used as input to the FiLM layers [37] in the conditional U-net, and $\texttt{pos\_emb}(k)$ is the positional embedding of the diffusion iteration $k$.

**SO(3)-equivariant conditional U-net.** A conditional U-net is composed of 1D convolution layers, upsampling layers, and FiLM layers. We make this network SO(3)-equivariant by converting every layer of this network to an SO(3)-equivariant layer.

To make 1D convolution layers SO(3)-equivariant, we treat vector channels of the layer inputs as batch dimensions and perform the original convolution operations. This simple change makes the convolution layer SO(3)-equivariant. We do not make any modifications to the upsampling layer, as it is naturally SO(3)-equivariant. To make the FiLM layer SO(3)-equivariant, we substitute vanilla linear layers with vectorized linear layers introduced in [14]. More formally, a FiLM layer is formulated as $\text{FiLM}(\mathbf{F}|\eta, \beta) = \eta\mathbf{F} + \beta$, where $\eta = f(\mathbf{x})$ and $\beta = h(\mathbf{x})$ are parameters predicted from learned functions used to modulate a neural network layer's activations $\mathbf{F}$, and $\mathbf{x}$ is this neural network layer's input. We replace non-equivariant layers $f$ and $h$ with 'vector neuron' layers [14], achieving rotation equivariance.

**Output.** The conditional U-net with SO(3)-equivariant layers processes $\mathbf{Z}_a$ and $\mathbf{Z}_c$ and outputs translation and scale invariant actions $\hat{\mathbf{A}}_{\text{inv}}$. To process this value into the final output of the policy, we assemble the final output action as $\hat{\mathbf{A}}_t = (\hat{\mathbf{A}}_{\text{inv}}^{(v)} \cdot \Theta_s, \hat{\mathbf{A}}_{\text{inv}}^{(d)}, \hat{\mathbf{A}}_{\text{inv}}^{(s)})$, where $\hat{\mathbf{A}}_{\text{inv}}^{(x)}$, $\hat{\mathbf{A}}_{\text{inv}}^{(d)}$, and $\hat{\mathbf{A}}_{\text{inv}}^{(s)}$ are the position, direction, and scalar components of the predicted invariant action.

Please refer to Section E in supplementary materials for implementation details.

## 4 Experiments

Through our experiments, we want to answer the following questions: (1) does our method generalize to unseen scenarios better than imitation learning methods that do not leverage equivariance; (2) does our method demonstrate more robust performance than prior methods for equivariant visuomotor policy learning that do not leverage diffusion models; (3) does our method achieve better data efficiency when there is limited training data; (4) how do different components of the method contribute to the final performance? We perform quantitative experiments in both simulations (Section 4.1) and the real world (Section 4.2) to answer these questions. Please view ablation experiments in Section B of the supplementary materials.

### 4.1 Simulation Experiments

#### 4.1.1 Comparisons to Vanilla Diffusion Policies and Other Equivariant Policy Architectures

Below, we evaluate our method on out-of-distribution generalization and compare to prior methods.

**Comparisons.** We compare our method to three baselines. (1) *Diffusion Policy (DP)* [8]: Vanilla diffusion policy using a point cloud as input; we substitute the imaged-based encoder to a PointNet++ encoder [39] similar to what EquiBot is using. (2) *Diffusion Policy with Augmentations (DP+Aug)*: This baseline uses the same architecture as the vanilla diffusion policy baseline, but trains with synthetically generated data augmentation. (3) *EquivAct* [16]: A re-implementation of [16] that drops the pre-training phase that requires task-specific simulated data.

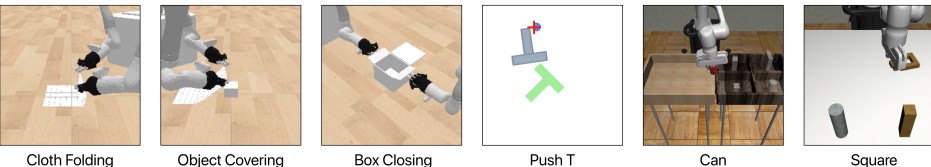

| Cloth Folding | Object Covering | Box Closing | Push T | Can | Square |

Figure 3: **Visualizations of simulation environments.** The three mobile manipulation tasks feature varied rigid, deformable, and articulated objects. The *Push T* task features multi-modal demonstration data that challenge the learning algorithms. The *Can* and *Square* tasks from the Robomimic benchmark require precise position and orientation movements to successfully complete the tasks.

**Augmentations.** The *DP+Aug* baseline requires augmentations in the training phase. In all four environments, we augment the training data to (1) rotate the observation around the z-axis, (2) uniformly scale the observation within the range $0.5 \times -1.5 \times$, and (3) apply a random Gaussian offset to the observation with standard deviation equal to $0.1$ times the approximate workspace size.

**Tasks.** We use four simulated tasks: *Cloth Folding*, *Object Covering*, *Box Closing*, and *Push T* (see Figure 3). The first three tasks involve two mobile robots manipulating various deformable and articulated objects. In these tasks, a simulated depth camera records point clouds of relevant objects in the scene from a third-person viewpoint. The policy takes these point clouds as input and commands the end-effector position, rotation, and gripper open-close actions of both robots. We train policies in these tasks with 50 synthetically generated demonstrations. The *Push T* benchmark task is a simulated 2D T-shape pushing game developed in [8] to showcase learning from multi-modal demonstrations. To make this task compatible with our setup, we assume the agent and object are placed on the ground plane ($z = 0$) in 3D space. In this task, the policy receives the eight corners of the T-shape as input and outputs the velocity command to the pushing operator. We use the same 200 demonstrations as [8] to train policies in this task.

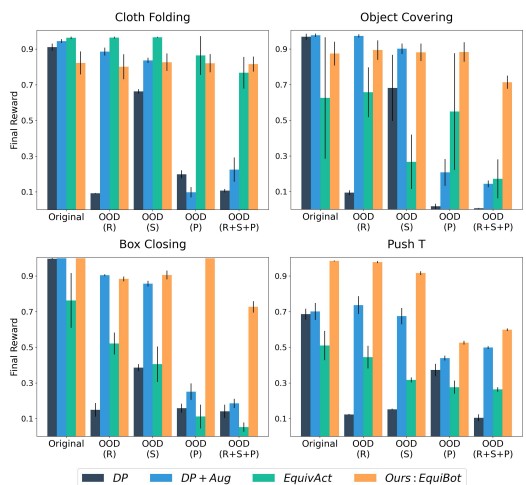

Figure 4: **Results of out-of-distribution generalization experiments.** We show that our method achieves more robust out-of-distribution generalization performance than methods that do not use diffusion processes to model policies and ones that do not utilize equivariance. Error bars show the mean and standard deviation over 5 checkpoints and 3 seeds.

**Training and evaluation.** We train all methods for 2,000 epochs on 3 random seeds. For every training run, we save a checkpoint every 50 epochs and evaluate the last 5 checkpoints saved at the end of training. For each evaluation, we run the policy in a randomly initialized environment for 10 episodes and record the mean final reward the policy achieves. This means that each bar in the resulting plot reports the mean and standard deviation of evaluation results over 3 seeds ×5 last checkpoints ×10 episodes = 150 trials of evaluation.

**Evaluation setups.** To gain insight into the generalizability of competing methods, we design four different evaluation setups to test our policies. The *Original* setup evaluates the policy at the same initial poses and goals as in the demos; the *OOD (R)* setup randomizes initial object rotation around the z-axis; the *OOD (S)* setup scales the scene up $1\times$ to $2\times$ with up to a 1.33 aspect ratio change; the *OOD (P)* setup adds dramatic position randomization to the scene; the *OOD (R+S+P)* combines previous OOD setups by randomizing rotation, scale, and translation of the scene.

**Results.** We show the results of this experiment in Figure 4. The *DP* baseline performs very well in the *Original* setup, but its performance drops significantly when it comes to any of the *OOD* setups. The *DP+Aug* baseline performs especially well in the *OOD (R)* setup because the augmentations

data this baseline learns from are in-distribution with respect to the evaluation scenarios. But in other scenarios such as *OOD (P)* and *OOD (R+S+P)*, it suffers from significant performance drops. The *EquivAct* baseline performs very well in the *Cloth Folding* task but displays subpar performance in *Object Covering* and *Box Closing* tasks. It also cannot perform well in the *Push T* task because the deterministic network architecture with behavior cloning loss cannot handle multi-modal training data well. We also show in supplementary materials Section B that *EquivAct* displays unstable training performance. Our method performs stably in all four tasks and suffers from the least amount of performance drop compared to all baselines. This shows that our method indeed outperforms prior methods in out-of-distribution generalization.

### 4.1.2 Data Efficiency Experiments

In this experiment, we aim to test if our method outperforms prior methods in a low-data regime, even if it is evaluated in distribution. Because we only care about in-distribution performance in this experiment, we only compare our method with the *DP* baseline. We adopt two Robomimic environments [15] for this experiment: *Can* and *Square*.

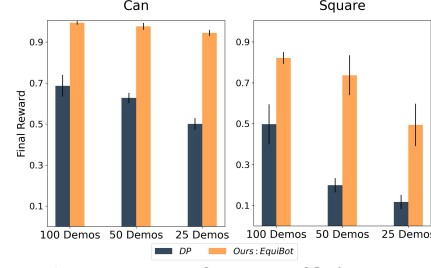

Figure 5: **Results of data efficiency experiments.** Our method achieves better data efficiency than the Diffusion Policy when evaluated in distribution on two benchmark tasks.

**Setup.** In each task, we train all methods 2,000 epochs in three setups: learning from 100 demos, 50 demos, and 25 demos. Following the same evaluation standard for mobile dual-robot and *Push T* environments, we report the mean and standard deviation over 150 trials of evaluation.

**Results.** The results of this experiment can be found in Figure 5. In both tasks, the performance of *DP* drops dramatically when the number of demos decreases from 100 to 25, while our method retains relatively higher performance when the amount of data drops. This is because when the training data size is too small to cover the whole distribution of initial poses, the *DP* baseline cannot naturally generalize to unseen initial poses during evaluation. Our method, on the other hand, leverages the equivariance nature to cope with novel initial object poses at test time.

## 4.2 Real Robot Experiments

We show a series of real robot experiments where we train mobile robots to perform everyday manipulation tasks from 5 minutes of single-view human demonstration videos. We select a suite of 6 tasks that involve diverse everyday objects, including rigid, articulated, and deformable objects (see Figure 6): (1) *Push Chair:* A robot pushes a chair towards a desk; (2) *Luggage Packing:* A robot picks up a pack of clothes and places it in an open suitcase; (3) *Luggage Closing:* A robot closes an open suitcase on the floor; (4) *Laundry Door Closing:* A robot pushes the door of a laundry machine to close it; (5) *Bimanual Folding:* Two robots collaboratively fold a piece of cloth on a couch; (6) *Bimanual Make Bed:* Two robots unfold a comforter to make it cover the bed completely.

**Data collection and robot setup.** We collect 15 human demonstration videos for each real robot task. We use a ZED 2 stereo camera to record the movement of a human operator using their fingers to manipulate the objects of interest at 15 Hz. After data collection, we use an off-the-shelf hand detection model [41], an object segmentation model [42], and a proprietary learned stereo-to-depth model to parse out the human hand poses and object point clouds in each frame of the collected demos. We then subsample this data to 3 Hz and convert it into a format supported by our policy training algorithm. In all real robot experiments, we use holonomic mobile bases [43] with Kinova Gen3 7 DoF arms mounted on top. Similar to human demonstration processing, we use a ZED2 camera and an object segmentation model to obtain the processed segmented point cloud as part of the input of our policy. See Section F of supplementary materials for more details.

**Training and evaluation.** We train all methods for 1,000 epochs. After training, we evaluate each method for 10 episodes and record the success rate of the method. We vary the evaluation

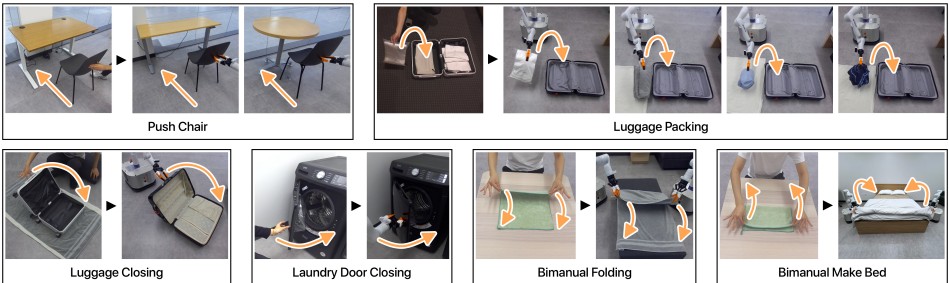

Figure 6: **Real robot evaluation setups.** Each block represents one task. In each block, we show sample training demos collected by a human on the left and evaluation scenarios on the right.

| Unseen Poses → Object Variations → | Push Chair – | | Laundry Door Closing | Luggage Closing – |
|---|---|---|---|---|
| | Long Desk | Round Table | – | Large Luggage |
| DP | 0/10 | 0/10 | 3/10 | 0/10 |
| DP+Aug | 1/10 | 2/10 | 2/10 | 2/10 |
| Ours | 8/10 | 10/10 | 8/10 | 6/10 |

| Unseen Poses → Object Variations → | Luggage Closing Translate + Rotate | | | | Bimanual Folding Translate + Rotate | Bimanual Make Bed – |
|---|---|---|---|---|---|---|
| | T-shirt | Towel Roll | Cap | Shorts | Long Bath Towel | Comforter |
| DP | 0/10 | 0/10 | 0/10 | 0/10 | 0/10 | 0/10 |
| Ours | 7/10 | 3/10 | 8/10 | 8/10 | 6/10 | 8/10 |

Table 1: **Results of real robot experiments.** In a suite of 6 mobile manipulation tasks, we show that our method can learn from just 5 minutes of human demonstration, outperforming the Diffusion Policy and the Diffusion Policy with Augmentation baselines by a large margin.

scenarios from the training scenarios differently in each task. In *Laundry Door Closing*, we perform evaluations in-distribution. In *Push Chair*, *Luggage Closing*, and *Bimanual Make Bed*, we evaluate with out-of-distribution objects. In *Luggage Packing* and *Bimanual Folding*, we not only switch to novel objects but also translate and rotate the layout of the scene.

**Results.** The results of real robot experiments are shown in Table 1. The evaluation shows that our method can generalize to diverse unseen objects, outperforming the *DP* baseline in both in-distribution and out-of-distribution scenarios with novel objects and unseen object poses. See supplementary materials Section C for more results and detailed analysis.

## 5   Conclusions, Limitations and Future work

We proposed EquiBot, a visuomotor policy learning method for generalizable and data-efficient policy learning in a wide range of robot manipulation tasks. In a suite of 6 simulated and 6 real robot tasks, we showed that our proposed method outperforms vanilla diffusion policies and prior imitation learning methods using equivariant architectures. We demonstrated that our method can learn from just 5 minutes of human demonstrations and generalize to unseen scenarios that are dramatically different from training scenarios.

While our method generalizes to scenes with unseen object positions, scales, and orientations, it does not handle nonlinear changes in object shapes or dynamics by construction. Our method also does not handle variations in the relative positioning of objects when multiple objects are present. Resolving these limitations might involve explicitly modeling scene dynamics and individual objects. Our method might also fail when the scene is partially occluded or the camera angle changes dramatically. This issue can be solved by learning general-purpose 3D representations robust to incomplete point clouds or novel viewing angles. Please see more discussions on limitations in Section H of supplementary materials. Solving the above challenges in out-of-distribution generalization and extending to multi-task setups are interesting directions for future work.

**Acknowledgments**

This work was supported in part by the Toyota Research Institute, Intrinsic and the Human-Centered AI Institute. The Stanford Robotics Center provided the space for our experiments.

We thank Jimmy Wu for helping with the hardware setup. We thank Xingze Dai and Yumeng Lu for helping with data collection. We thank Cheng Chi and Yilun Du for sharing insights about Diffusion Policy. We thank Yihuai Gao and Haoyu Xiong for helping with the fin ray gripper setup.

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

# A Further Discussion of Related Works

**Imitation learning from 3D inputs.** While many imitation learning works assume RGB images as visual observations [15, 4, 20, 8], some works, similar to the design choice of our work, assume 3D point clouds or depth inputs to their methods. Recently, Ze et al. [44] proposed *3D Diffusion Policy*, a depth-only variant of diffusion policy for visuomotor policy learning. Their method is very similar to our *DP* baseline, with two differences: (1) they use a simpler *DP3 encoder* in their work; (2) they use a 2-layer MLP to encode the robot proprioceptive states before concatenating with the point cloud representation.

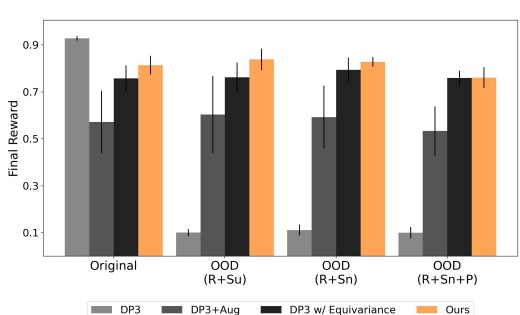

In Figure 7, we show quantitative comparison results between our method and baselines related to [44]. As in our main paper results, we run 3 seeds of training runs, each for 2,000 epochs, for all experiments. We evaluate the last 5 checkpoints for each training run and collect the final task reward for 10 episodes in each evaluation. Comparisons show that [44] displays similar performance as the *DP* baseline in the main paper, performing very well in in-distribution setups and poorly in out-of-distribution setups.

We also show that [44] can be easily integrated with our method by switching our PointNet-based encoder to the *DP3 encoder*. In Figure 7, we show a comparison between our method and a variation of our method with a modification of the *DP3 encoder* to make it SO(3)-equivariant. Results show that the DP3 variant has slightly lower but comparable performance as our method in the *Cloth Folding* task.

Figure 7: **Comparisons with DP3-related architectures in the Cloth Folding task.** We compare our method with two DP3-related baselines and one variation of our method that uses the DP3 encoder: (1) *DP3* is a variation of the *DP* baseline with the PointNet-based encoder replaced by the *DP3 encoder* proposed in [44], with the code of *DP3 encoder* copied verbatim from the public codebase; (2) *DP3+Aug* is a variant of the DP3 baseline trained with augmentations that are the same as the *DP+Aug* baseline in the main paper; (3) *DP3 w/ Equivariance* is the integration of DP3 into our method.

**Additional works in equivariant architectures for robot manipulation.** Aside from prior equivariant architectures for robot manipulation introduced in the main paper, there are also robotics works that attempt to utilize equivariance in various setups. Some works [45, 46] attempt to use equivariance in a pick-and-place setup, while others [47, 48] propose SO(2)-equivariant robot policies for tabletop manipulation tasks. In comparison to prior works, our proposed architecture is equivariant to position, orientation, and uniform scaling. In addition, our method can be applied in various 3D manipulation tasks that involve rigid, deformable, and articulated objects.

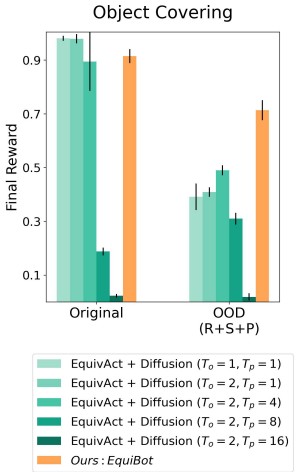

# B Ablations and Analysis

**Ablations on architecture designs.** We perform more ablation experiments to understand how various aspects of our proposed method contribute to the final performance of the method. The first question we want to answer is how much the architecture of our method matters compared to that of the prior work *EquivAct* [16]. To show this, we provide a comparison of EquiBot with a naive extension of *EquivAct* with a diffusion head. As seen in Figure 8, EquiBot significantly outperforms this architecture for all hyperparameter variations we tried in the OOD setups. When naively adding

Figure 8: **Ablations on model architecture.** EquiBot outperforms naive extensions of EquivAct [16] for a variety of hyperparameter choices.

a diffusion head to *EquivAct* with action horizon 1, the policy's poor performance can be explained by its inability to generate multiple actions in a sequence to ensure temporal action consistency [8]. When we increase prediction horizons, the *EquivAct* architecture, which is not designed to deal with large prediction horizons, suffers from dramatic drops in performance. Our method performs the best across all comparisons we performed in Figure 8.

**Ablations on equivariance.** To show how different forms of equivariance play a role in the final performance of our method, we perform ablation experiments in the *Object Covering* task by subtracting rotation, translation, and scaling equivariance implementations from our method. The result of this ablation experiment is shown in Figure 9. As shown in the results, when we subtract out any form of equivariance from the implementation, the performance of this ablated method suffers from performance drops when the evaluation setup is out-of-distribution to the equivariance this method supports.

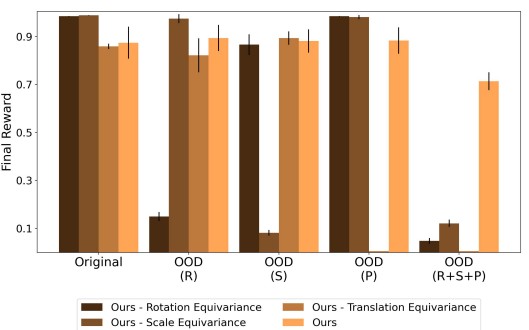

Figure 9: **Ablations on equivariance in object covering.** We show that translation, rotation, and scale equivariance each contribute to our method.

**Training stability.** In the real world, it is impractical to test multiple training checkpoints on the robot. Therefore, high training stability can make checkpoint selection and getting robust test-time performance easy. Here, we specifically compare the training stability of our method with prior work *EquivAct* [16]. In Figure 10, we plot the in-distribution performance of EquivAct against our method in the Push T task during the training process. We plot average reward over 40 evaluation episodes for every 50 epochs of training. Our method achieves more stable training performance across checkpoints than EquivAct.

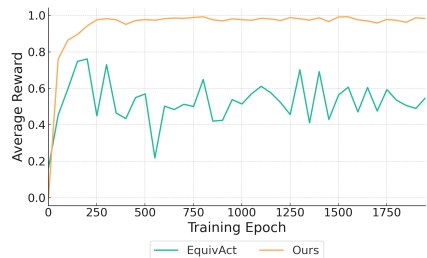

Figure 10: **Training stability.**

## C Further Evaluations and Performance Analysis

### C.1 Failure Analysis

Although our method outperforms vanilla Diffusion Policy [8] and prior equivariant visuomotor policy architectures, our method still presents various failure cases. Below, we focus our analysis on execution failure of our method in the real robot experiments. In Figure 11, we show the failure breakdown of all real robot executions we have performed with our method.

In most packing tasks (packing t-shirt, towel, and cap), the main failure cases are the end-effector opening too early. We believe this is because the out-of-distribution scenarios resulted in the agent thinking that it has moved to the dropping location for the object and opened the end-effector. For the packing shorts task variation, half of the failures come from end-effector opening too late, and half of the failures come from the shorts not slipping off from the end-effector due to the friction of the end-effector. This problem can potentially be solved by designing end-effectors that can handle deformable objects better or performing online adaptation after training, which is out of the scope of our work.

In the *Push Chair*, *Laundry Door Closing*, and *Bimanual Folding* tasks, the majority of failures come from the end-effector not performing the full motion or not performing gripper open close actions at the right time. This most likely happens because the errors in predicted actions accumulated and the observation became too out-of-distribution scenarios for the policy to behave correctly. The *Bimanual Make Bed* task appears to be more difficult for our object segmentation method than other

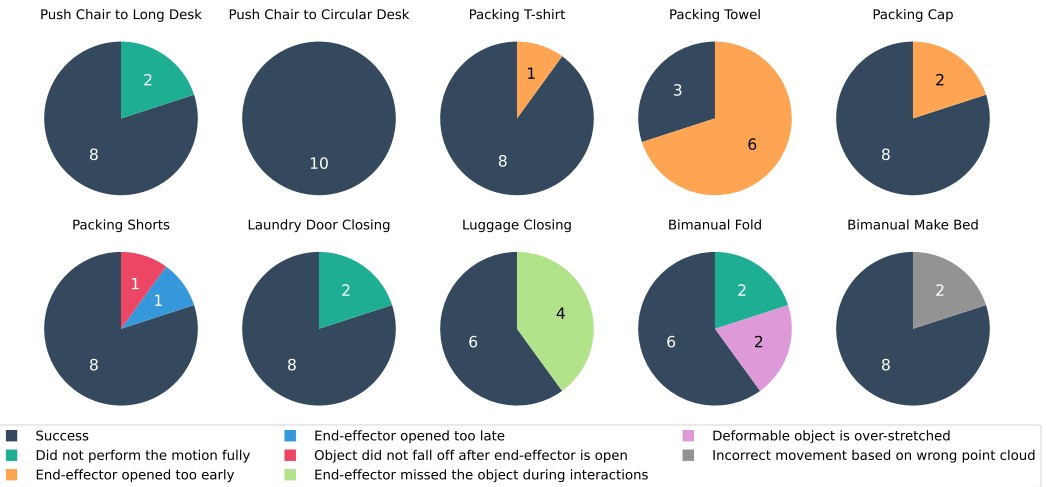

Figure 11: **Failure cases of our method during real robot executions.** The pie charts show failure breakdown in every real robot task variation. The navy color denotes success, while other colors denote different types of failures. Each pie chart shoes a total of 10 trials since we run 10 episodes per evaluation.

|  | **Ours** | | | **DP** | | |
|---|---|---|---|---|---|---|
| **Number of Demos →** | 10 | 25 | 50 | 10 | 25 | 50 |
| Success Rate | 8/10 | 9/10 | 10/10 | 3/10 | 5/10 | 7/10 |
| Close with Click Sound | 2/10 | 2/10 | 5/10 | 2/10 | 1/10 | 5/10 |
| Total Missing Angle | 17.46° | 7.18° | 6.3° | 140.42° | 33.85° | 21.55° |
| Collision or Safety Issue | 0/10 | 0/10 | 0/10 | 2/10 | 0/10 | 0/10 |

Table 2: Detailed performance of the laundry door closing task.

tasks, causing failures to segment the full comforter in some scenarios, since the folded comforter looks like two pieces of cloth, one laid on top of another.

## C.2    Additional Real Robot Results

**Detailed performance analysis of the laundry door closing task.** To understand the performance of the Diffusion Policy [8] and our method better, we perform a more detailed analysis in the laundry door closing task. In Table 2, we report four different metrics of policy performance in each evaluation setup. *Success Rate* measures the percentage of evaluations that end within the success criteria we set; *Close with Click Sound* measures the percentage of episodes that end with the laundry door closed completely after making a clicking sound; *Total Accumulated Missing Angle* measures the sum of the opening angles of the laundry door at the end of the 10 evaluation episodes; *Collision or Safety Issue* measures the percentage of evaluation runs that are terminated because of undesired collisions or critical safety issues, such as the robot arm getting stuck at the laundry door. From the evaluation results, we see that the *DP* policy not only has a lower success rate but also has a much larger accumulated missing angle. The baseline also suffers from many safety issues requiring episodes to be manually terminated by the robot operator. Our method has much fewer safety issues when it executes.

**Qualitative results.** In Figure 12, we show qualitative rollout samples for all evaluation scenarios we mentioned in the paper, plus one bonus task where two robots lift a woven basket onto a coffee table.

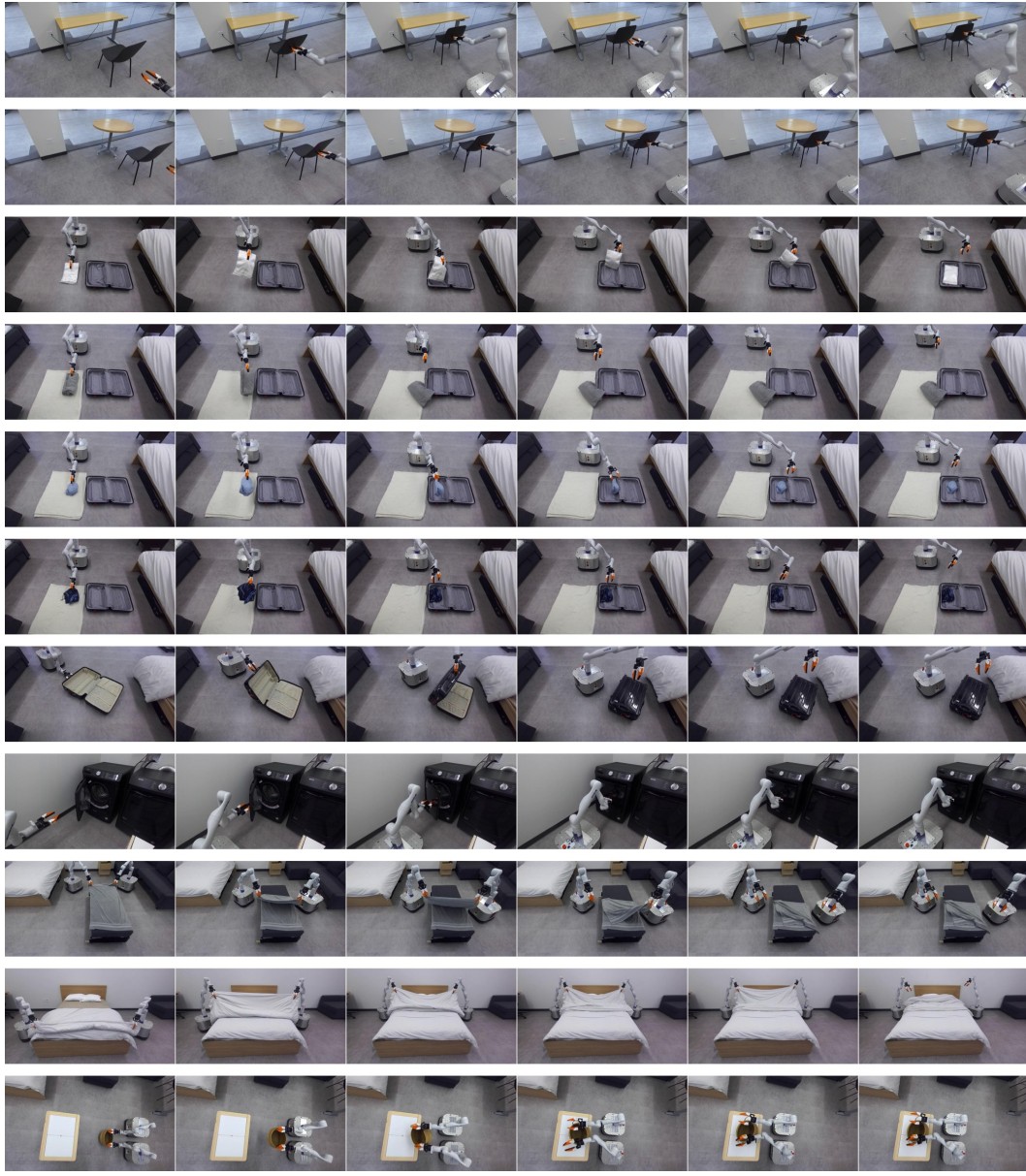

Figure 12: **Qualitative rollout samples for all real robot evaluation scenarios.** From top to bottom, we have: (1) pushing a chair towards a long standing desk; (2) pushing a chair towards a circular table; (3) packing t-shirts; (4) packing towel roll; (5) packing cap; (6) packing shorts; (7) closing a check-in luggage; (8) closing laundry door; (9) bimanual folding; (10) bimanual make bed; (11) a bonus task where two robots lift a woven basket onto a coffee table.

## D  Equivariant Distributions and Diffusion

In this section, we prove Proposition 1 that we introduced in Section 3.2 of the main paper.

**Proposition 1.** *Let $p(x^K|c)$ be a SO(3)-equivariant density function conditioned on c, i.e. $\forall \mathbf{R} \in SO(3), p(x^K|c) = p(\mathbf{R}x^K|\mathbf{R}c)$. If Markov transitions $p(x^{k-1}|x^k, c)$ are SO(3)-equivariant for all k, i.e. $p(x^{k-1}|x^k, c) = p(\mathbf{R}x^{k-1}|\mathbf{R}x^k, \mathbf{R}c)$, then the density $p(x^0|c) = \int p(x^T|c)\Pi_{k=1}^K p(x^{k-1}|x^k, c)$ is also SO(3)-equivariant.*

*Proof.* We say that a distribution $p(y)$ invariant to the SO(3)-rotation group actions if:

$$p(y) = p(\mathbf{R}y), \quad \forall \mathbf{R} \in \mathrm{SO}(3). \tag{4}$$

We say that a conditional distribution $p(y|x)$ is equivariant to SO(3) rotations if:

$$p(y|x) = p(\mathbf{R}y|\mathbf{R}x), \quad \forall \mathbf{R} \in \mathrm{SO}(3). \tag{5}$$

[49] shows that an invariant distribution composed with an equivariant invertible function results in an invariant distribution. Moreover, given a Markov chain $\boldsymbol{x}^{0:K}$, [50] shows that if the initial distribution $x^K \sim p(x^K)$ is invariant to a group and the transition probabilities $x^{k-1} \sim p(x^{k-1}|x^k)$ are equivariant at each time step to the same group, then the marginal distribution of $x^{k-1}$ is also invariant to the group actions at each time step. Specifically, $p(x^0)$ is invariant:

$$p(x^0) = \int p(x^K) p(\boldsymbol{x}^{0:K-1}|x^K) \mathrm{d}\boldsymbol{x}^{1:K} \tag{6}$$

$$= \int p(x^K) \prod_{k=1}^{K} p(x^{k-1}|x^k) \mathrm{d}\boldsymbol{x}^{1:K} \tag{7}$$

$$= \int p(\mathbf{R}x^K) \prod_{k=1}^{K} p(\mathbf{R}x^{k-1}|\mathbf{R}x^k) \mathrm{d}\boldsymbol{x}^{1:K} \tag{8}$$

$$= p(\mathbf{R}x^0). \tag{9}$$

Now consider an additional condition $c$, and equivariant initial distribution $p(x^K|c)$ with transitions $p(x^{k-1}|x^k, c)$ as follows:

$$p(x^K|c) = p(\mathbf{R}x^K|\mathbf{R}c), \quad p(x^{k-1}|x^k, c) = p(\mathbf{R}x^{k-1}|\mathbf{R}x^k, \mathbf{R}c). \tag{10}$$

The following shows that the marginal distribution $p(x^0|c)$ is also equivariant:

$$p(x^0|c) = \int p(x^K|c) p(\boldsymbol{x}^{0:K-1}|x^K, c) \mathrm{dx}^{1:K} \tag{11}$$

$$= \int p(x^K|c) \prod_{k=1}^{K} p(x^{k-1}|x^k, c) \mathrm{d}\boldsymbol{x}^{1:K} \tag{12}$$

$$= \int p(\mathbf{R}x^K|\mathbf{R}c) \prod_{k=1}^{K} p(\mathbf{R}x^{k-1}|\mathbf{R}x^k, \mathbf{R}c) \mathrm{d}\boldsymbol{x}^{1:K} \tag{13}$$

$$= p(\mathbf{R}x^0|\mathbf{R}c), \tag{14}$$

$\square$

## E   Method Architectures and Implementation Detail

In this section, we describe in detail the architecture of our method. We visualize the architecture of our model in Figure 13.

**Observation and action spaces.** In all simulated and real robot tasks except for *Push T*, we use a 13-dimensional proprioception information and a 7-dimensional action space for each robot. The proprioception data for each robot consists of the following information: a 3-dimensional end-effector position, a 6-dimensional vector denoting end-effector orientation (represented by two columns of the end-effector rotation matrix), a 3-dimensional vector indicating the direction of gravity, and a scalar that represents the degree to which the gripper is opened. The action space for each robot consists of the following information: a 3-dimensional vector for the end-effector position velocity, a 3-dimensional vector for the end-effector angular velocity in axis-angle format, and a scalar denoting the gripper action.

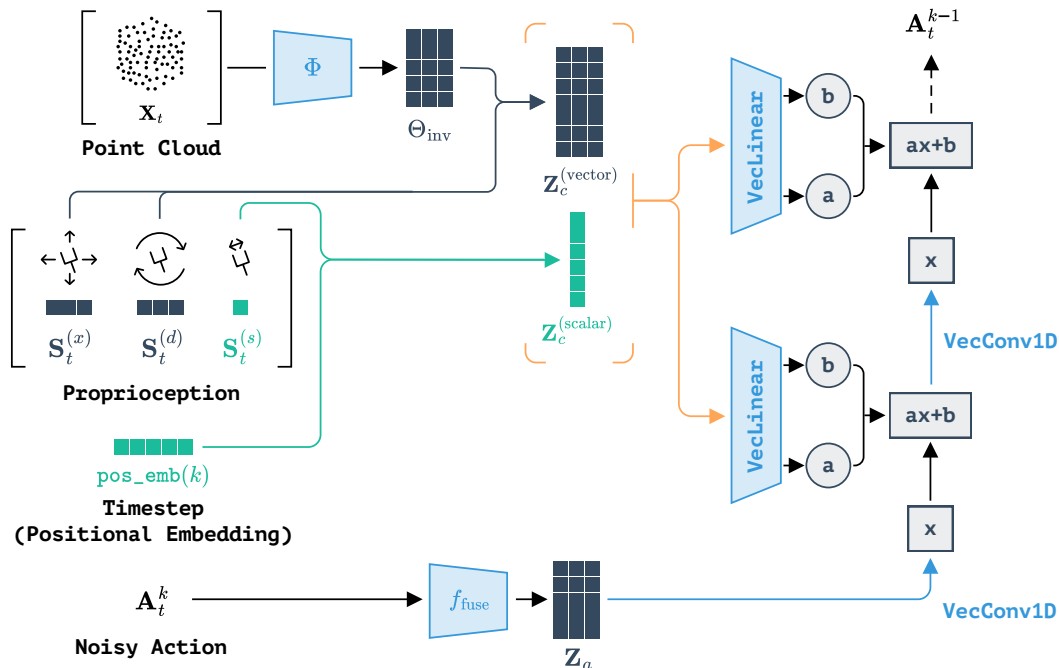

Figure 13: **Architecture of EquiBot.** Given input scene point cloud, robot proprioception, noisy actions, and the diffusion timestamp, our architecture processes position, direction, and scalar information independently, uses the encoder outputs to scale position information into position and scale invariant values, and then routes them into an SO(3)-equivariant conditional U-net to predict denoised actions. In the figure, we omit scaling for the ease of viewing. `VecLinear` and `VecConv1D` refer to a SO(3)-equivariant version of linear and convolution 1D layers.

In the *Push T* task, the robot proprioception is 3-dimensional and consists of the agent's 3D position in the scene, while the action space is 3-dimensional and denotes the absolute position target of the agent.

In all simulated and real robot tasks, our policy uses an observation horizon of 2 steps, a prediction horizon of 16 steps, and an action horizon of 8 steps. This is identical to the setup used in the diffusion policy paper [8].

**Encoder architecture.** In all tasks except for *Push T*, we use a SIM(3)-equivariant version of PointNet++ with 4 layers and hidden dimensionality 128. In the *Push T* task, we decrease the number of layers to 2 since the number of points in the point cloud observation is much smaller in this task.

**Noise prediction network.** Our noise prediction network inherits hyperparameters from the original diffusion policy paper [8]. In all simulation experiments, we use the DDPM scheduler [13] and perform 100 denoising steps during inference. In real robot experiments, to optimize for inference speed, we use the DDIM scheduler [51] with 8 denoising steps.

**Point cloud size.** Picking the number of points to sample in the point cloud observation is a key hyperparameter to consider when designing an architecture that takes point cloud inputs. In our experiments, we found out that using 512 or 1024 points is sufficient for all tasks. In particular, for all real robot experiments and simulated mobile manipulation tasks, we use 1024-point point clouds. In *Can* and *Square* tasks, we use 256 and 512 points respectively since there is relatively more training data in these tasks, and decreasing the number of points in the point cloud makes training faster without hurting performance.

**Normalization.** Data normalization can be important to the performance of diffusion models. Vanilla diffusion policy normalizes the observations and actions separately. We instead normalize all 3D-vector inputs (including observation and action) together due to our SIM(3)-equivariance

assumptions. Implementation-wise, we take a subset of training data and compute the mean point cloud scale and mean action scale as $s_{\mathrm{pc}}$ and $s_{\mathrm{ac}}$. Then, all position- and velocity-related information is divided by $s_{\mathrm{pc}}/s_{\mathrm{ac}}$ at the start of the network forward pass and multiplied back before the output is returned. This normalization factor ensures that the diffusion process always works with actions with values within the $-1$ to $1$ range. We normalize scalar information in the same way as the vanilla diffusion policy.

# F    Real Robot Setup and Experiments – Further Details

## F.1    Human Demo Parsing Infrastructure

We use a single ZED 2 camera to record human natural motion in real-time, which is way more flexible and time-efficient than the expert demonstration from human teleoperation of a robot. With that, we create a set of human demonstrations $\mathcal{D} = \{\tau^n\}_{n=1}^{N}$. Each human demonstration consists of a series of RGB-D image frames $\tau^n = \{I_t^n\}_{t=1}^{T}$, where $T$ is the episode horizon.

The human demonstration processing module has three parts:

1. an off-the-shelf object detection and tracking model $\mathbf{X}_t^n = \Psi(I_t^n, I_{t-1}^n)$ that takes the current and previous demonstration frames $I_t^n$ and $I_{t-1}^n$ as input, and outputs a parsed point cloud of objects of interest $\mathbf{X}_t^n$;
2. an off-the-shelf hand detection model $\mathbf{H}_t^n = \Phi(I_t^n)$ that takes the current demonstration frame as input and outputs the keypoints on each human hand in the input frame $\mathbf{H}_t^n$;
3. an alignment module $\Omega(\mathbf{X}_t^n, \mathbf{H}_t^n, I_t^n)$ that takes the outputs of the previous two steps as input, aligns the 3D coordinates of the outputs, and outputs the aligned human finger pose $\mathbf{y}_t^n$ in the same coordinate system of $\mathbf{X}_t^n$.

We use Grounded Segment Anything Model with DEVA [52] as the object detection and tracking model $\Psi$ and HaMeR [41] as the hand detection model $\Phi$. In the alignment module, we find a set of matching points between the point cloud $\mathbf{X}_t^n$ and $\mathbf{H}_t^n$, and then fit a rotation transformation $R_h$ and an offset $t_h$ to transform all points $\mathbf{H}_t^n$ to the coordinate frame of the point cloud. Then, we extract the thumb and index finger positions on the transformed keypoint set to predict the human "end-effector" pose $\mathbf{y}_t^n$. We found that the alignment module $\Omega$ is crucial, as the hand detection model produces hand poses in a different coordinate frame from the point cloud. Without this module, the resulting hand poses will not align with the object point cloud.

## F.2    Mobile Robot Control Infrastructure

Our robot control setup consists of a centralized workstation and two mobile robots. The workstation reads and parses visual observations, performs policy inference, and communicates with the robots. The mobile robots take the output actions of the policy and execute them. On the workstation side, we build a multi-process infrastructure to handle observation parsing, policy inference, and action execution.

In the observation parsing process, we obtain visual observations from a single ZED 2 camera directly connected to the workstation via cable. We use the Grounded Segment Anything Model with DEVA [52] to obtain segmented point clouds that contain only relevant objects in the scene. We then downsample this segmented point cloud to 1024 points. The downsampled point cloud and the robot proprioception information are sent to the policy inference process. The policy inference process then outputs a sequence of 16 predicted actions.

In the action execution process, we first reset all robots to their initial poses. Then, for each step in an evaluation episode, we read out the latest policy inference results from the policy inference process. To ensure accurate execution, we compute the elapsed time between the policy input time and action execution time. If this elapsed time exceeds a threshold, we skip the first few predicted actions during action execution. After skipping the first few predicted actions to account for latency,

we select the 8 actions that immediately follow the skipped actions to execute. This means that no matter how many actions are skipped, we always execute 8 actions at a time.

On the mobile robot, we also build a multi-process control infrastructure to control the Kinova arm and mobile base. We utilize a motion capture system to obtain mobile base position. Then, we (1) transfer each control signal from the global world frame to the local frame of the Kinova arm, (2) convert the target pose at the gripper fingertip to an expected pose of the Kinova end-effector, and (3) convert it into a velocity command for the arm. We use position control for the mobile base and only move it when the robot end-effector is too close to or too far from the base.

### F.3 Task Details

**Push chair.** In this task, the human demonstrations are collected using a standing desk ($48 \times 30$ inches). The policies are evaluated on two different tables: a longer rectangular desk ($58 \times 23$ inches) and a circular table (diameter of 36 inches). An episode is considered successful if the center of the chair goes beneath the desk.

**Luggage packing.** In the human demonstrations, a human picks up a pack of white t-shirts and places them into a white carry-on luggage. At evaluation time, we test four different packing items: white t-shirts (same as training object), gray towel roll, blue cap, and navy shorts. An episode is considered successful if at least half of the packed object ends up within the luggage.

**Luggage closing.** In this task, human demonstrations are collected on a small carry-on luggage ($55 \times 40 \times 23$ cm), while the policies are evaluated on a large check-in luggage ($76 \times 48 \times 25$ cm). An episode is considered successful if the luggage ends up in a closed state.

**Laundry door closing.** In this task, the human demonstrations and the robot work with the same laundry machine (front-loader). The goal is to close the door of the laundry machine that is open at the start of the episode. An episode is considered successful if the door ends up with an opening of at most 5cm.

**Bimanual folding.** In this task, the human demonstrations are collected by using two hands to fold a small piece of cloth ($34 \times 38$ cm). At evaluation time, the robot is asked to fold a large gray towel ($140 \times 75$ cm). After each evaluation episode, we measure the mean distance between each grasped corner to their corresponding target cloth corners and mark the episode as successful if this mean distance is less than 0.2 times the length of the folding side of the cloth.

**Bimanual make bed.** In this task, the human demonstrations are collected by using two hands to unfold a towel ($34 \times 38$ cm). At evaluation time, the robot is asked to make the bed by unfolding a much larger comforter on top of the bed. After each evaluation episode, we measure the mean distance between each grasped corner to the bed headboard and mark the episode as successful if this mean distance is less than 0.2 times the length of the bed.

## G Simulation Tasks – Further Details

**Cloth folding.** In this task, the demonstrations show two robots folding a piece of cloth ($27.5 \times 27.5$ cm). During an evaluation, we compute the task reward as $1.0 - (d_1 + d_2)/(0.275 \times 2)$, where $d_1$ and $d_2$ denote the distance from the two grasped cloth corners to the target cloth corners.

**Object covering.** In this task, the demonstrations show two robots moving a piece of cloth ($27.5 \times 27.5$ cm) onto a rigid box ($10 \times 7 \times 5$ cm). During an evaluation, the task reward is computed as $V_{\text{intersect}}/V_{\text{convex hull}}$, where $V_{\text{intersect}}$ is the volume intersection between the box and the convex hull of the cloth and $V_{\text{convex hull}}$ is the volume of the convex hull of the cloth.

**Box closing.** In this task, the demonstrations show two robots closing a box ($14.5 \times 12 \times 11.5$ cm) with three flaps. Success in this task is evaluated as $(a_1 + a_2 + a_3)/(3 \times 180)$, where $a_1$ to $a_3$ denote the angle in degrees at which each flap of the box is closed.

**Push T.** In this task, a 2D anchor pushes a T-shaped object on a plane of dimension $512 \times 512$ pixels. The task reward is computed as the percentage of the T shape that overlaps with the target T pose.

**Robomimic tasks.** We use the same object and reward specifications in these tasks as the original benchmark. Please check out the Robomimic [15] paper for more details.

## H  Limitations and Future Work – Further Details

Although we only enforced equivariance to SIM(3)-transformations by construction, in practice we still observed generalization to geometric variations beyond SIM(3), such as non-uniform scaling. While this capability is not enabled by construction, our hypothesis is that ensuring equivariance in all policy layers is conducive to learning a more general feature representation. This may relate to the observations in prior feature-learning works, e.g. [53], which noted that by training the network to be (in their case) invariant to data transformations, they gained better features for downstream perception tasks. In future work, it would be interesting to study the intermediate features of equivariant networks with systematic probing and evaluation techniques.

