# OpenReview forum: "EquiBot: SIM(3)-Equivariant Diffusion Policy for Generalizable and Data Efficient Learning"
_robot-learning.org/CoRL/2024/Conference — CoRL 2024_

### Official Review · Reviewer_vrP9 · 2024-07-08
**The presented policy achieves SIM(3) equivariance, but only to a limited degree.**

**Originality:** 2
**Technical Quality:** 3
**Clarity Of Presentation:** 4
**Potential Impact:** 3
**Recommendation:** 3
**Confidence:** 5

**Review:**

quality

Overall the paper achieves SIM(3) equivariance, but the method is constrained by canonicalization. Experiments showed the efficacy of the method, but ablation experiment is missing.

clarity

The presentation is clear, except that the Figure.3 in Appendix does not agree with Equation.1, 2 in the main text.

originality

This work has marginal novelty, because SE(3) equivariant diffusion planner has been explored previous work [1]. The contribution could be more significant if the method is applied to a broader manipulation tasks.

significance of this work

-strengths

Experiments demonstrate the generalization and sample efficiency by leveraging SIM(3) equivariance.

-weaknesses

Achieving SIM(3) equivariance by canonicalization limited the expressiveness of the method. Moreover, the method is based on type1 irreps in the feature, which potentially limits the rotational accuracy of the policy. In contrast to canonicalization, group convolution or higher types of irreducible representation of SO(3) could be utilized to increase expressiveness.

Lastly, there is no ablation experiment.

[1] J. Brehmer, J. Bose, P. De Haan, and T. S. Cohen. Edgi: Equivariant diffusion for planning with embodied agents. Advances in Neural Information Processing Systems, 36, 2024.

**Quality Of The Limitations Section:**

3

**Questions For Rebuttal:**

Does the proposed method outperform the baseline (Diffusion Policy) on broader class of tasks? I.e., coffee preparation in the MimicGen that includes multiple objects and long horizon reasoning?

Could author provide ablation experiments that show amount the SIM(3) (rotation, translation, and scale) equivariance, which equivariance is most critical?

Could author provide experiments similar to “out-of-distribution generalization experiments” that test the model on rotation, translation, and scale out-of-distribution individually?

Section. Equivariant diffusion architectures in the Appendix seems the most related works, could you put it in the main text?

**Robotics Focus:**

4

**Summary Of Paper:**

This work proposes a SIM(3) (translation, rotation, scale) equivariant diffusion policy. The translation and scale equivariance are achieved by canonicalization. The rotation equivariance is based on type1 irreps (vector). First the VN point net++ extract type1 irreps, then a modified FiLM conditioned temporal Unet achieves SO(3) equivariance, due to 3 copies of standard Unet are used to process the XYZ- components of the type1 irreps independently. Processing XYZ- components of type1 irreps is also a form of canonicalization.  The method achieves strong generalization to novel SIM(3) transformed scenes as well as sample efficiency for a specific class of tasks. However, due to the limitation of canonicalization, the method do not generalize to broader tasks that involving muti-objects.

**Summary Of Recommendation:**

This work leverages SIM(3) equivariance in the diffusion policy by using cannonicalization. However, the method is limited to simple tasks, i.e., when there is only one target object in the scene.

---

### Official Review · Reviewer_U9p1 · 2024-07-19
**EquiBot demonstrates impressive real-world results but questionable simulation results.**

**Originality:** 2
**Technical Quality:** 3
**Clarity Of Presentation:** 3
**Potential Impact:** 3
**Recommendation:** 2
**Confidence:** 5

**Review:**

**Strengths:**
- The real world experiments are fairly impressive with complex realistic tasks being solved using the proposed method.
- Paper is well written and easy to understand/follow.
- Scale invariance is typically overlooked in equivariant models and is included in this work.
- This work is generally valuable. Data efficiency is a big challenge in robotics especially in these complex mobile robot manipulation domains where expert demonstrations are expensive. Equivariant models has been shown in both this and other works to be effective at addressing this.

**Weaknesses:**
- The primary weakness of this work is the performance of the baselines. Namely the performance of the diffusion policy baseline is lower than reported in the original Diffusion Policy work. While some of this could be explained by the change from RGB-D observations to point clouds, the authors state that they use the key point observations for the Push-T task and report significantly worse performance for the diffusion baseline than in the original work. This makes it difficult to trust the baseline performance for the other tasks as it implies some issue w/the baseline. This might be explained by the amount of expert data used for training but this is not stated anywhere as far as I see.
- Additionally, when benchmarking in simulation more than 10 episodes should be used to evaluate the methods.
- Limitations section is a bit bare-bones. Would be interesting to discuss the effect of incomplete point clouds on the method as this is a common failing of SO(3) equivaraint models.
- As this builds on top of EquiAct, the paper assumes a decent amount of pre-knowledge about the workings of EquiAct encoders.
- Novelty is a bit limited as this is primarily an extension to EquivAct where a diffusion head is placed onto of the EquiAct backbone.

**Quality Of The Limitations Section:**

3

**Questions For Rebuttal:**

1. Can the authors provide more detail on the simulation experiments to explain the poor performance of diffusion when compared to that in the original work?
2. Additional evaluation episodes for simulation experiments would strengthen the claims.
3. How many demonstrations were used in the simulation experiments?
4. Is our understanding of this method being an extension of EquiAct with a diffusion policy head correct?

**Robotics Focus:**

4

**Summary Of Paper:**

This paper propses EquiBot, an extension of EquiAct to close-loop visuomotor control tasks. By combining SO(3) equivariant models with diffusion the authors show they can solve several robot manipulation tasks in simulation and in the real world.

**Summary Of Recommendation:**

This paper proposes a extension of a previous work (EquiAct) to close-loop control and is therefor somewhat incremental. The real world experiments are very impressive but the baseline results in simulation seem worse than previously reported.

---

### Official Review · Reviewer_8ciL · 2024-07-21
**Initial Review**

**Originality:** 2
**Technical Quality:** 3
**Clarity Of Presentation:** 2
**Potential Impact:** 2
**Recommendation:** 3
**Confidence:** 4

**Review:**

Overall the problem setting investigated in this paper is very important as equivariant / invariant policies have the potential to significantly improve data-efficiency which is crucial in imitation learning. However, for me, the relation and contributions w.r.t. the prior work of Equivact is unclear and needs further clarification. Moreover, I found the paper rather difficult to read and follow.

In the following, I provide a more detailed review.

1) Introduction. Overall, I am missing more citations in the introduction. While the introduction in its current form nicely motivates the direction of the work I feel that one single citation for an entire intriduction section is simply not sufficient given the vast amount of work in imitation learning, imitation learning for manipulation, and also in policy representation. The paper definitely needs more evidence (in the form of citations) to underline the authors claims. Moreover (but this is just a minor comment), I felt that the sentence in line 26-27 a bit too much of an overstatement, i.e., ".... that depart drastically from the scenarios in which it is trained". While I fully agree with the authors that equivariance is a strong property and can circumvent already quite some variations that would otherwise cause problems. Nevertheless, there exist way more environment variations where equivariance alone is not sufficient. Thus I suggest to tone down the statement.

2) Related work and in particular the relation with the other work [17] is not entirely clear and should be further clarified. Is it right that this work here essentially shares the same encoder as [17] and only uses a different policy backbone / policy parameterisation? If yes, this should be written and explained more clearly. Moreover, it would be essential that the authors provide more explanation and also evidence w.r.t. the fact that "... we show that this method displays unstable training performance" (line 72). As far as I see, at the moment the paper and supplementary only shows that empirically during evaluation the performance is different, however, I would argue that different performance is not necessarily related with "unstable training performance".

3) "Assumptions to observation and action spaces" at the end of Section 3.1. Since the representation of the observation and action spaces is essential, additional information here would significantly improve the paper. In particular, what is the magnitude of the end effector orientation that are then expressed as velocity vectors? Why are end effector positions integrated and translated into a set of points instead of using the velocity representation? Also, how exactly is the action representation retrieved / converted to desired end effector poses that the robot should go through? In particular, the authors mention the executed actions consist of 3D offsets or velocities, normalized directions and scalars. In my opinion more details are required here. As I see, there are also no details provided on this in the Appendix which makes it difficult to fully understand the method. This also continues in the later part (Section 3.2.) of the paper, where the sentence in lines 140-142 of how the "relevant part of the action" is scaled back was not understandable for me. I therefore suggest to define how actions are encoded, processed in the architecture, updated, etc. more strictly, in the form of equations.
On top of that I wonder, when the iterative denoising process is run, do you have to go multiple times through the Point-net based encoder, and if yes, what is the computational complexity of the method, i.e., what is its runtime?

4) Moreover (again rather minorily), I feel the statement "... other forms of visual information do not have the necessary 3D information to make it equivariant to translation, rotation, and scaling" (lines 122-124) a bit too strong. Combined with proprioception / pose information of a camera I think it might also be possible to integrate such observations into equivariant policies. I do agree that this might be less straightforward but still possible. Thus I would suggest to tone down this statement.

5) In the simulation experiments to improve readability, I propose to put the augmentations before the tasks, I feel that this would improve readability.

6) In the experiment results, I did not understand what is meant by the fact that the Equivact policy cannot handle multi-modal training data well (line 251). I assume this is related with the behavior cloning loss employed in EquivAct w.r.t. the DiffusionPolicy employed in this work. However, it would help to write this explicitly here in the paper.

7) To potentially save space, I propose to merge the Data collection paragraph and the Mobile robot setup as many parts are repeated and it is also inconsistent as in the Mobile robot setup there are the citations that are missing in the Data collection paragraph.

In light of the thorough rebuttal provided by the authors, I raise my score to weak accept. However, I want to point out that I agree with the other reviewers in that the paper would benefit significantly if also results were shown on more complex tasks (including more than one object) or long horizon tasks.

**Quality Of The Limitations Section:**

3

**Questions For Rebuttal:**

- Please explain in more detail how this work is different from [17]
- What exactly do you mean in unstable training performance of [17] w.r.t. to this work?
- How exactly are the actions encoded in the representation and how is the delta in the actions received as output and used to update the actions?
- Why are end effector positions integrated and translated into a set of points instead of using the velocity representation?
- For the iterative denoising process, does the computation time scale linear with the number of denoising iterations? Also, how does this compare with the computational requirements with [17]?

**Robotics Focus:**

4

**Summary Of Paper:**

In this paper, the authors present a new SIM(3)-equivariant diffusion policy that by definition is invariant w.r.t. environment changes in scale, rotation and translation. The authors evaluate the method's efficiency across simulated and real robot experiments.

**Summary Of Recommendation:**

Overall I think the paper tackles an interesting and important problem. However, in its current form it lacks clarity in multiple locations, the authors need to better highlight the difference w.r.t. the very related work [17], better stress the contributions of this work, and better explain what they mean with training stability.

---

### Author Rebuttal · Authors · 2024-08-11

The attached PDF contains (1) plots for additional experiments we ran in the rebuttal phase on page 1 and (2) the revised paper starting from page 2. We have highlighted the changes we made in the revised paper.

---

### Decision · Program_Chairs · 2024-09-04

**Decision:**

Accept

**Comment:**

The paper introduces a SIM(3)-equivariant diffusion policy, which aims to maintain invariance to changes in scale, rotation, and translation in robotic environments. The reviewers acknowledge the importance and potential impact of equivariant policies for enhancing data efficiency in imitation learning. However, concerns are raised about the clarity of the paper, particularly in relation to prior work (EquiAct) and the detailed methodology, which requires further explanation and citations. Reviewers also noted the necessity for additional experiments, including ablation studies and broader task evaluations, to fully validate the approach. Despite these issues, the real-world experiments and the inclusion of scale invariance are appreciated.
During the rebuttal period, although there was not much discussion between the authors and the reviewers, there was a discussion amongst the reviewers. All agreed that the rebuttal had lead to a better draft of the paper, and most concerns were addressed. There were however concerns regarding the performance gap of diffusion policy in pushT between this paper and the original diffusion policy paper.